# Effects of fenbendazole on fecal microbiome in BPH/5 mice, a model of hypertension and obesity, a brief report

Kalie F. Beckers[1], Christopher J. Schulz[2], Chin-Chi Liu[1], Elise D. Barras[1], Gary W. Childers[2], Rhett W. Stout[1], Jenny L. Sones[1,3]*

1 Veterinary Clinical Sciences, School of Veterinary Medicine, Louisiana State University, Baton Rouge, Louisiana, United States of America, 2 Department of Biological Sciences, Southeastern Louisiana University, Hammond, Louisiana, United States of America, 3 Pennington Biomedical Research Center, Louisiana State University, Baton Rouge, Louisiana, United States of America

* jsones@lsu.edu

## Abstract

Fenbendazole (FBZ) is a common antiparasitic treatment used in research rodent colonies for biosecurity purposes. The effect of this compound has been studied in C57 mice, but never before in a strain of mice that has co-morbidities, such as the blood pressure high (BPH)/5. The BPH/5 mouse is an inbred genetic model of hypertension. While both male and female BPH/5 have high blood pressure, there is a metabolic sexual dimorphism with females displaying key features of obesity. The obese gut microbiome has been linked to hypertension. Therefore, we hypothesized that fenbendazole treatment will alter the gut microbiome in hypertensive mice in a sex dependent manner. To test the influence of FBZ on the BPH/5 gut microbiota, fecal samples were collected pre- and post-treatment from adult BPH/5 mice (males and non-pregnant females). The mice were treated with fenbendazole impregnated feed for five weeks. Post-treatment feces were collected at the end of the treatment period and DNA was extracted, and the V4 region of 16S rRNA was amplified and sequenced using the Illumina MiSeq system. The purpose was to analyze the fecal microbiome before and after FBZ treatment, the results demonstrate changes with treatment in a sex dependent manner. More specifically, differences in community composition were detected in BPH/5 non-pregnant female and males using Bray-Curtis dissimilarity as a measure of beta-diversity (treatment p = 0.002). The ratio of Firmicutes to Bacteroidetes, which has been identified in cases of obesity, was not altered. Yet, Verrucomicrobia was increased in BPH/5 males and females post-treatment and was significantly different by sex (treatment p = 5.85e-05, sex p = 0.0151, and interaction p = 0.045), while Actinobacteria was decreased in the post-treatment mice (treatment p = 0.00017, sex p = 0.5, interaction p = 0.2). These results are indicative of gut dysbiosis compared to pre-treatment controls. *Lactobacillus* was decreased with FBZ treatment in BPH/5 females only. In conclusion, fenbendazole does alter the gut microbial communities, most notable in the male rather than female BPH/5 mouse. This provides evidence that caution should be taken when providing any gut altering treatments before or during mouse experiments.

**Data Availability Statement:** All raw sequence reads, and corresponding metadata can be found on the SRA website accession number PRJNA851094.

**Funding:** National Institutes of Health (NIH) P20GM135002; Louisiana State University Veterinary Clinical Sciences CORP grant. The funders had no role in study design, data collection and analysis, decision to publish, or preparation of the manuscript.

**Competing interests:** The authors have declared that no competing interests exist.

**Abbreviations:** FBZ, Fenbendazole; BPH, Blood pressure high; ASV, amplicon sequence variant; PERMANOVA, Permutational multivariate analysis of variance; FDR, False Discovery Rate; clr, centered log ratios.

## Introduction

Fenbendazole (FBZ) is a broad spectrum benzimidazole anthelmintic that is commonly used in laboratory animals for the treatment of pinworms [1]. The drug is regarded as safe, including use during pregnancy, and has minimal side effects [1]. FBZ impregnated feed is the current treatment of choice to eliminate rodent internal parasites. While poorly absorbed, bioavailability is enhanced when taken with a meal [2]. Benzimidazoles act by altering microtubule function through inhibition of microtubule polymerization and binding to B-tubulin [3]. Benzimidazoles have been shown to have adulticidal, larvicidal, and ovicidal properties [4,5]. Studies in rodents have indicated multiple physiologic effects of FBZ, including changes in immune parameters [6–8], but more research is needed to evaluate the effects of FBZ on the gut microbiome in a sex dependent manner in mice with co-morbidities, which are phenotypic hallmarks of genetic mouse models.

The treatment used in this study was FBZ impregnated Purina 5001 chow fed ad libitum for five weeks. The treatment duration considers the lifecycle of the parasites as well as the persistence of the parasite eggs in the environment. Pinworms (order *Oxyurina*) are one of the most common parasites found in modern rodent facilities [9–11]. Typically in mice, pinworm infestations are subclinical, but heavy burdens can lead to rectal prolapse, enteritis, and intussusception, especially in strains with co-morbidities [12–14]. Decreased weight gain, diminished growth rates, increased caloric demands, and overall nutritional status can be compromised [12,15,16]. Parasite load can also be influenced by the animals age, weight, strain/stock and sex [10,11,13]. As a result, infections may influence experimental outcomes.

The ideal gut microbiome is rich and diverse. Changes in the gut microbiome are referred to as a dysbiosis. Dysbiosis can lead to adverse health outcomes. This study aims to evaluate the changes in the gut microbiome pre- and post-five-week treatment of FBZ in the BPH/5 (blood pressure high subline 5) mouse model. The BPH/5 mice demonstrate cardiometabolic disease spontaneously [17,18]. This mouse model has been used to study preeclampsia (PE) [18–22]. Because this mouse spontaneously develops PE, increased blood pressure during pregnancy, we can investigate pre-pregnancy risk factors. The BPH/5 females have elevations in blood pressure and obesity compared to C57 females before pregnancy [23]. The BPH/5 males demonstrate increased blood pressure without obesity compared to C57 males unlike the BPH/5 females [24]. The overarching hypothesis is that FBZ treatment will alter the gut microbiome in a sex dependent manner in BPH/5 mice. This study will compare the microbial communities of the BPH/5 male and female pre- and post-treatment to evaluate the effect of FBZ on the gut microbiome on a mouse strain with coupled comorbidities.

## Materials and methods

### Animal experiments

This is a cross-over study evaluating the effects of fenbendazole (FBZ) on the gut microbiome in a sex dependent manner. Adult (age range: 8 weeks-6 months of age) female and male BPH/5 mice were used in this study. BPH/5 mice were a gift from Dr. Robin Davisson, Cornell University. They are maintained as an in-house colony at Louisiana State University. Mice were housed in Micro-Isolator™ cages (Lab Products, LLC. Aberdeen, Maryland, USA) on individually ventilated cage racks (Lab Products, LLC. Aberdeen, Maryland, USA). The cages are placed in a temperature- and humidity-controlled facility, maintained on a 12-hour light/dark cycle, and fed standard mouse chow (LabDiet 5001 Laboratory Rodent Diet, LabDiet, St. Louis, MO, USA) with water available ad libitum. The mice were co-housed and randomly selected for sampling. Two female mice were lost before follow up and one male pre-treatment

sample had insufficient fecal material collected. Two groups of mice were used: 1) BPH/5 pre- and post- treatment males (n = 9 and n = 10, respectively) and 2) BPH/5 pre- and post- treatment non-pregnant females (n = 12 and n = 10, respectively). The pre-treatment mouse served as its own control in this study. All procedures were approved by Louisiana State University's Institutional Animal Care and Use Committee.

**Treatment.** Mice were fed FBZ impregnated feed (Mod LabDiet 5001 w/150ppm Fenbendazole, LabDiet, St. Lous, MO, USA) ad libitum for five weeks. Complete cage changes occurred every two weeks in which feed was completely changed out during this time. Feed was topped off as needed in between cage changes. Cages (including the box, lid, cage card holder, feed hopper, shelter, water bottle, and water bottle lids) were cleaned via a cagewasher (NorthStar Model R630, BetterBuilt, Delta, BC, Canada) every two weeks. Cages are washed for 25 minutes using Uri-Solve (Pharmacal Research Laboratories, Inc. Waterbury, CT, USA) and Clout (Pharmacal Research Laboratories, Inc. Waterbury, CT, USA) cleaning agents with a final rinse temperature of 180˚F. Personal Protective Equipment (PPE) was donned and doffed prior to entering and exiting the room, respectively. PPE included shoe covers, a disposable gown, nitrile gloves, and a hair net.

**Samples collection.** Fecal samples were collected pre-treatment and at the end of the five weeks post treatment. To collect feces, mice were placed individually in sterile empty cages and allowed to defecate voluntarily. Samples were placed in sterile tubes and stored at –80˚C until further analysis.

**DNA sequencing.** Microbial DNA was extracted from fecal samples using the Qiagen DNeasy PowerSoil extraction kits (Qiagen, USA) according to manufacturer's protocol, with the initial step of the protocol beginning with vertexing the sample with beads to break up the material. The V4 variable region of the 16S rRNA gene was amplified with PCR primers 515f/806r [25] in a 30 cycle PCR using the DreamTaq Hot Start PCR Master Mix Kit (Thermoscieniftic, Waltham, MA). PCR was performed in 20 µl vol and included: 2 µl (7.5 µM concn) of forward and reverse primers, 12.5 µl of Hot Start Taq 2X Master Mix (New England BioLabs Inc., Ipswich, MA., USA), 3.5 µl of deionized water, and 2 µl of sample DNA. Thermal cycle conditions were 95˚C for 3 min for initial denaturing step, followed by 30 cycles of 95˚C for 30 s, 50˚C for 1 min, and 72˚C for 1 min. PCR products were checked on a 2% agarose gel for correct product size formation (approx 350 bp). Michigan State University Genomics Core performed library preparation prior to Illumina MiSeq sequencing following the manufacturer's guidelines [25,26]. Reagent controls using certified DNA free water were run through library preparation and PCR and did not generate libraries. For quality control, samples submitted for sequencing included a random blank sample of technical replicates.

**Bioinformatics.** Initial quality screening, demultiplexing, amplicon sequence variant (ASV) inference and chimera removal were performed using the DADA2 package [27]. ASVs were classified using the Silva Release 132 16S rRNA database [28,29]. Microbial community analysis (Alpha and Beta Diversity) was performed using the vegan R package [30]. Shannon diversity index was used to analyze alpha diversity, while Bray-Curtis dissimilarity was used to examine beta diversity. Permutational multivariate analysis of variance (PERMANOVA) [31] was performed using vegan package Adonis function. Statistical analysis was performed using GraphPad Prism Version 9.5.0 for Windows (GraphPad Software Inc., San Diego, CA). To determine differentially abundant ASVs, the ASV table was first trimmed to only include ASVs with an abundance of two or more ASVs across all samples. Two-way ANOVAs were performed on the trimmed ASV table using centered log ratio transformed abundances with time, sex, time x sex as the fixed effects and each mouse as the random effect [32]. Shannon diversity index against with treatment, gender and their interactions. ANOVA p-values were corrected using the False Discovery Rate (FDR) method of Benjamini and Hochberg [33]. All

figures were presented as means± SEM. P values and FDR-corrected p values <0.05 were considered significant. All raw sequence reads and corresponding metadata can be found on the SRA website accession number PRJNA851094.

## Results

### Changes at the phylum level in BPH/5 mice with Fenbendazole treatment

Of the 40 samples processed and sequenced, all achieved a threshold above 10,000 reads with an average of 76,524 per sample after quality filtering. A bar graph was used to visualize the relative abundance at the phylum level within each group (Fig 1). The top phyla consisted of Firmicutes, Bacteroidetes, Epsilonbacteraeota, and Proteobacteria. The BPH/5 males had an increase relative abundance in Firmicutes from 39.44% to 49.67% with a decrease in Bacteroidetes of 48.40% to 36.81% post-treatment (Fig 2A). Epsilonbacteraeota remained similar with 7.44% and 7.96% pre- and post-treatment, similarly Proteobacteria and Cyanobacteria did not vary with 1.74% and 1.86%, and 2.12% and 1.86% pre and post, respectively (Fig 2A). Tenericutes decreased from 0.053% to undetected and Verrucomicrobia increased from undetected to 1.64% (Fig 2A). The BPH/5 females showed more similar relative abundances pre- and post-treatment (Fig 2B). A slight decrease in Firmicutes from 47.11% to 44.7%, Bacteroidetes remained similar 41.17% and 39.51%, slight increase in Epsilonbacteraeota 6.61% to 10.65%, Proteobacteria remained unchanged 2.55% and 2.51% (Fig 2B). Cyanobacteria was not affected 2.17% and 2.21%, Verrucomicrobia nor Tenericutes were either, 0.00% and 0.17%, 0.01% and 0.00%, respectively (Fig 2B). The Firmicutes to Bacteroidetes ratio, which is an indicator of obesity in humans [34], was not significantly different between groups (Fig 2C). In addition, the centered log ratios (clr, log base 2) of the different phyla were assessed to

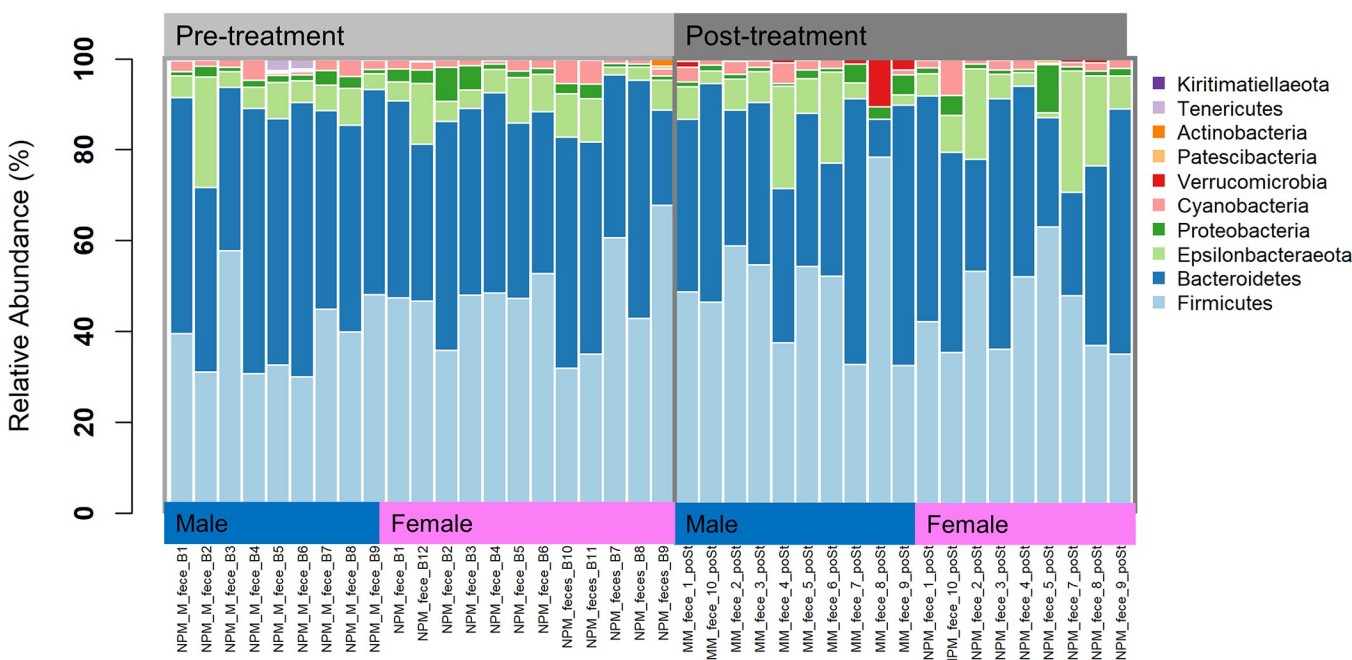

**Fig 1. Bar graph of the relative abundance at the phyla level of BPH5 mice with fenbendazole treatment.**

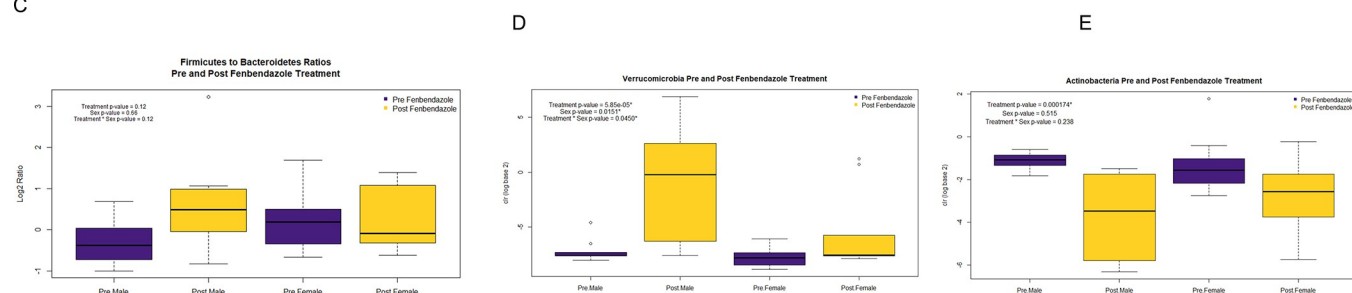

**Fig 2. Differences at the phyla level of BPH5 male and females pre and post fenbendazole treatment.** A). Relative abundance at the phyla level of the BPH5 male pre and post-treatment. B). Relative abundance at the phyla level of the BPH5 female pre and post-treatment. C). Ratio of Firmicutes to Bacteroidetes in BPH/5 male and females. D). Changes in Verrucomicrobia pre and post fenbendazole treatment in male and female BPH/5 mice. E). Changes in Actinobacteria pre and post fenbendazole treatment in male and female BPH/5 mice.

standardize the abundances of the microbes. Verrucomicrobia increased post-treatment and had significant differences between sex and interaction between sex and treatment (treatment p = 5.85e-05, sex p = 0.0151, and interaction p = 0.045), (Fig 2D). This demonstrated that the treatment had an effect in a sex dependent manner. Actinobacteria significantly decreased post-treatment but did not have differences by sex or interaction (treatment p = 0.00017, sex p = 0.5, interaction p = 0.2). (Fig 2E).

## Microbial community composition of the BPH/5 mouse with Fenbendazole treatment

To assess changes in gut microbial diversity with treatment Shannon's index was assessed. Diversity significantly decreased after FBZ treatment, but no difference was observed between sex or interaction between treatment and sex (treatment p = 0.02, sex p = 0.19, interaction p = 0.17) (Fig 3A). Since observed diversity can be affected by sampling effort, i.e., sequencing depth, samples were rarefied to 5000 observations per sample randomly with 1000 trials to assess the significance of treatment on Shannon's index. Treatment was significant (p < 0.05) for all rarefied samplings with a range of 0.005 to 0.014. To determine if community composition was different between treatments (beta diversity), a Bray-Curtis dissimilarity matrix was calculated from the relative abundances of ASVs in samples. A principal coordinate analysis (PCoA) plot was created using the Bray-Curtis dissimilarity (Fig 3B). Separation between pre- and post- treatment samples was observed along the first two principal coordinates which accounted for 19.33% and 13.12% of the variance between the samples. Treatment had a significant effect on community composition using PERMANOVA of the Bray-Curtis dissimilarity matrix, with both BPH/5 female and male microbial communities differing post-treatment but not different by sex or interaction (treatment p = 0.002, sex p = 0.3, interaction p = 0.05) (Fig 3B).

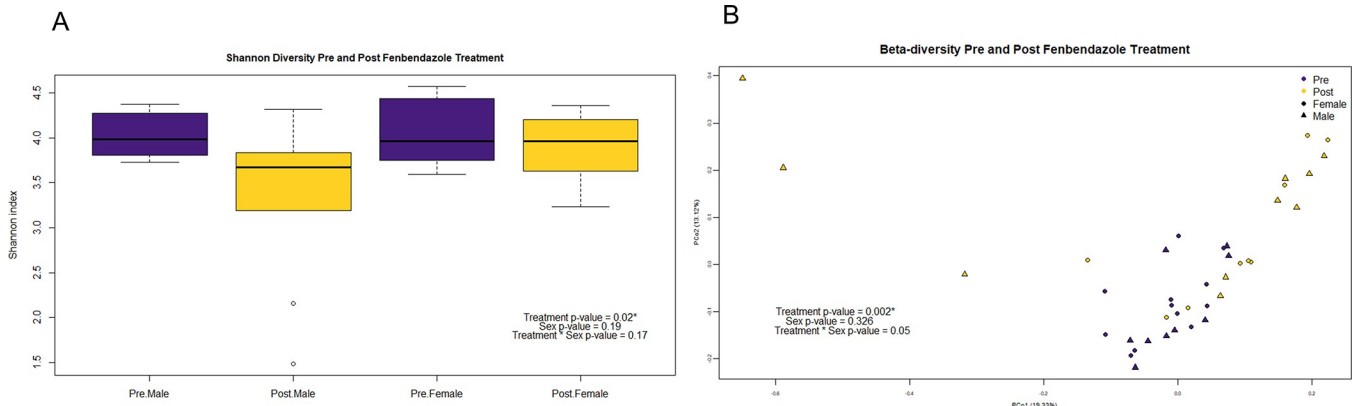

**Fig 3.** A.) Alpha diversity of BPH5 mice treated with Fenbendazole B.) Principal coordinate analysis of Bray-Curtis dissimilarity matrix of BPH5 mice treated with Fenbendazole.

## Changes at the ASV level with Fenbendazole treatment and phenotypic outcomes

Looking further there were changes at the ASV level using the centered log ratios (clr, log base 2) of the relative abundance mentioned above, the BPH/5 males had a relative decrease in *Lachnoclostridium spp*, *Lachnospiraceae spp*, *Odoribacter spp*, *Clostridiales family Xiii ssp*, *Alistipes spp*, and *Muribaculceae ssp* (Fig 4A). While in the BPH/5 females, *Lactobacillus spp*, *Ruminococcus_1 ssp*, and *Muribaculaceae spp* were all found to be decreased with FBZ treatment (Fig 4B). Since the phylum Verrucomicrobia was found to be significantly different by treatment, sex, and interaction it was investigated further. *Akkermansia* was the found at the genus level of the phyla Verrucomicrobia, *Akkermansia* (Verrucomicorbia) was significantly increased in males post treatment and was greater in males compared to females (Fig 2). Within the phylum Actinobacteria which was significantly reduced with treatment in both sexes, *Enterorhabdus* was the major genus that contributed to these changes.

## Discussion

Routine veterinary care in mouse vivariums is necessary for colony maintenance. Herein we discovered that FBZ does alter the gut microbial communities when analyzing alpha and beta-diversity in adult BPH/5 females and males. The most notable changes were seen in the BPH/5 males where the change in diversity is seen at the phylum level. It has been well documented that FBZ treatment has no effect on reproduction or food intake in rodents [35]. Therefore, even though FBZ is safe to use it could be altering the gut microbiome and changing experimental results in animals with co-morbidities.

A study by Korte et al, evaluated the gut microbiome on C57 female mice using FBZ impregnated feed and topical moxidectin quarantine protocol and found that time or location played a larger role in alterations of the gut microbiota than quarantine treatments on its composition [36]. A notable difference between that study and ours is that we looked at both sexes, not just female mice. In addition, we used the BPH/5 mouse which harbors comorbidities, hypertension and predisposition for obesity in females.

First, phylum change in fecal communities was analyzed after FBZ in both male and female BPH/5. While there were statistical differences in Firmicutes and Bacteroidetes abundance in BPH/5 males after FBZ, this was not observed in females. Moreover, the ratio of Firmicutes to Bacteroidetes was not altered in either BPH/5 males nor females, who display hallmark obesity

A

## Top 6 changes at ASV level in BPH5 males with Fenbendazole treatment

| ASV # | Name | Change | P value |
|---|---|---|---|
| 312 | Lachnoclostridium spp | ↓ | 2.74e-5 |
| 170 | Lachnospiraceae spp | ↓ | 5.21e-4 |
| 55 | Odoribacter spp | ↓ | 5.35e-4 |
| 288 | Clostridiales family Xiii | ↓ | 6.99e-4 |
| 121 | Alistipes spp | ↓ | 8.74e-4 |
| 240 | Muribaculaceae spp | ↓ | 1.71e-3 |

B

## Top 6 changes at ASV level in BPH5 females with Fenbendazole treatment

| ASV # | Name | Change | P value |
|---|---|---|---|
| 4 | Lactobacillus spp | ↓ | 6.9e-5 |
| 65 | Ruminococcus_1 spp | ↓ | 1.79e-4 |
| 199 | Muribaculaceae spp | ↓ | 1.24e-3 |
| 43 | Lactobacillus spp | ↓ | 1.53e-3 |
| 240 | Muribaculaceae spp | ↓ | 2.26e-3 |
| 113 | Muribaculaceae spp | ↓ | 4.74e-3 |

**Fig 4.** Top 6 changes at ASV level in a sex dependent manner A). Changes found with fenbendazole treatment at the ASV level in BPH5 male. B). Changes found with fenbendazole treatment at the ASV level in BPH5 female.

phenotypes [23,37]. Furthermore, their body weights during the 5-week treatment period (average body weight: male 25.26 ± 0.4180g and female 23.17± 0.4792) were similar to historical age-matched BPH/5 body weights [17,23]. Alterations in their gut microbiome due to treatment may have downstream effects of gut-derived metabolites. The gut microbiome influences short chain fatty acids that effect the overall energy homeostasis and cardiovascular health and disease [38,39]. This is an area that is currently under investigation in the laboratory. Next, Verrucomicrobia and Actinobacteria were significantly altered with FBZ treatment. Verrucomicrobia was increased and was significantly different by sex and treatment interaction. This demonstrates that the treatment had an effect in a sex dependent manner. In a similar study by Gorla et al, Verrucomicrobia increased with the use of an antiparasitic agent [40], which was mirrored in our study. Additionally, Actinobacteria was another phylum that was found to decrease post-treatment using the clr method in this study. In a study by He et al, Actinobacteria decreased with the treatment of FBZ in the Amur tiger [41].

When investigating Verrucomicrobia at the genus level, *Akkermansia* was found in high abundances. *Akkermansia* (Verrucomicorbia) was significantly increased in males post treatment and was greater in males compared to females (Fig 2). Denoting the sex specific change at the genus level. In humans, *Akkermansia* is involved in glucose metabolism and obesity [42]. *Enterorhabdus* from the phylum Actinobacteria was overrepresented and reduced post treatment in both sexes. *Enterorhabdus* is related to inflammation and obesity in other mice strains [43,44].

A healthy gut is characterized by high diversity [45]. Alpha diversity which is a measure of evenness and richness was decreased post FBZ in both male and female BPH/5 fecal communities. Thus, demonstrating a potential microbial dysbiosis with FBZ in BPH/5 mice. Beta diversity which is a measure of community structure was altered with FBZ in BPH/5 mice and this emphasizes that treatment potentially causes a microbial community shift.

The BPH/5 females showed a marked decrease in *Muribaculaceae spp*.,but more clinically relevant is the profound decrease in *Lactobacillus* found in the BPH/5 female post-treatment. *Lactobacillus* is a known genusthat promotes healthy intestinal function and reduces inflammation [46–48]. In future microbiome investigations in BPH/5, it will be important to measure Lactobacillus as a potential biomarker of cardiometabolic health and disease. A limitation of this study was not directly measuring the effects of FBZ on blood pressure or obesity in the mice sampled for this study. However, in house colony mice used for blood pressure studies before [18] and post FBZ [17,24], still exhibited high mean arterial pressure. This suggests that treatment did not change the hypertensive phenotype in these mice and their hallmark comorbidity was maintained.

Although FBZ is not typically considered antibacterial, it has been shown to influence bacterial antimicrobial resistance [49]. Changes in the gut microbiome can affect experimental outcomes(5). In this study, male mice demonstrate decreases of *Lachnoclostridium spp*, *Lachnospiraceae spp*, *Odoribacter spp*, *Clostridiales family Xiii ssp*, *Alistipes spp*, and *Muribaculceae ssp*. The important contributors here are *Clostridiales family Xiii ssp*, *Alistipes spp*, and *Muribaculceae ssp*. *Clostridiales* family has been previously described to decrease with the use of antibiotics [50] and also decreases with the onset of colitis [51]. *Muribaculceae* has also been shown to decrease in abundance with the development of colitis [52]. While, *Alistipes* shows protective attributes in colitis, cancer, and cardiovascular disease [53]. These are main contributors found in this study that may implicate a gut microbiome shifting towards dysbiosis.

In conclusion, this study presents that there are changes in the gut microbial communities after FBZ treatment. These results were different by sex, herein because *Akkermansia* (Verrucomicrobia) had a greater effect on the males post treatment. Significant changes at the ASV level were also different by sex. Additionally, 16S metagenomic sequencing is relatively insensitive when compared to a complete genomic analysis. Although, the microbial alterations should be a consideration for investigators if FBZ treatment is warranted during experiments.

## Author Contributions

**Conceptualization:** Kalie F. Beckers, Jenny L. Sones.

**Data curation:** Kalie F. Beckers, Christopher J. Schulz, Elise D. Barras, Rhett W. Stout, Jenny L. Sones.

**Formal analysis:** Kalie F. Beckers, Christopher J. Schulz.

**Funding acquisition:** Jenny L. Sones.

**Investigation:** Kalie F. Beckers, Jenny L. Sones.

**Methodology:** Kalie F. Beckers, Chin-Chi Liu, Jenny L. Sones.

**Supervision:** Jenny L. Sones.

**Writing – original draft:** Kalie F. Beckers.

**Writing – review & editing:** Christopher J. Schulz, Chin-Chi Liu, Elise D. Barras, Gary W. Childers, Rhett W. Stout, Jenny L. Sones.

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
