## [Decision Letter · Decision Letter 0]

31 Jan 2023

PONE-D-23-00107Effects of fenbendazole on fecal microbiome in BPH/5 mice, a model of hypertension and obesity, a brief reportPLOS ONE

Dear Dr. Sones,

Thank you for submitting your manuscript to PLOS ONE. After careful consideration, we feel that it has merit but does not fully meet PLOS ONE’s publication criteria as it currently stands. Therefore, we invite you to submit a revised version of the manuscript that addresses the points raised during the review process. Both reviewers found the study to be of value; however, additional clarification of the results is necessary. Furthermore, the discussion and conclusions should be edited to match the data generated in this study, as indicated by the reviewers.

We look forward to receiving your revised manuscript.

Kind regards,

Christopher Staley, Ph.D.

Academic Editor

PLOS ONE

“Yes

JLS

NIH (P20GM135002)”

“NO”

Reviewers' comments:

Reviewer's Responses to Questions

**Comments to the Author**

1. Is the manuscript technically sound, and do the data support the conclusions?

Reviewer #1: Partly

Reviewer #2: Yes

2. Has the statistical analysis been performed appropriately and rigorously? 

Reviewer #1: Yes

Reviewer #2: Yes

3. Have the authors made all data underlying the findings in their manuscript fully available?

Reviewer #1: Yes

Reviewer #2: Yes

4. Is the manuscript presented in an intelligible fashion and written in standard English?

Reviewer #1: No

Reviewer #2: Yes

5. Review Comments to the Author

Reviewer #1: Abstract: line 42, define FBZ.

Please clearly mention which comparisons (gender, medication, before/after medication, obesity) are you targeting. Each result line in mentioned for a different comparison. Please give

Give Bray-Curtis dissimilarity p value. The difference is only between male and female which means treatment has no effect. Please give each p values. Results are not coherent with the conclusions.

I wouldn’t say Lactobacillus as a marker of healthy gut. Lactobacillus at genus level has more complex role.

Methods:

What we mean by Adult (8 weeks-6 months of age)?

Line 86 to 91: break in smaller sentences.

Line 102, 103: were not are

Line 107 : exiting the room, respectively

Line 109: Samples collection:

Line 110: individually in sterile empty cages

Line 101: were placed in sterile tubes

Line 129: Microbial community

Changes at the phyla level in BPH/5 mice with Fenbendazole treatment

Herein just mention treatment effects (FBZ). Or if you want to add gender as well, make a crystal clear difference from the start.

Only mention changes with p value < 0.05 or elsewhere mention p value. Result sentences are not written properly, and each sentence has multiple results with no clear reflection. Sometimes past and present are mixed (164-165).

Either figures are not clear, or p values are not mentioned. For figure 2. C, D, and E, use the same headings. As I can see that the post-male has larger bell size, this may lead to non-significant results. Lines 181 to 184: reflect that the effect is due to treatment which contradicts with the conclusions. Overall, I believe that we shall rewrite results with more coherent text and figures.

179: dissimilarity

Discussion:

Line 203: What we mean by institution?

Figure 2 A and B; also use the similar titles. In PCoA I can see that post treatment clustering is poor, which needs to be justified.

Reviewer #2: You should acknowledge as a limitation that although you found some differences in microbial frequency, you did not evaluate effects on blood pressure or obesity, which are key features of the animal model you studied. Also, 16S rRNA testing is relatively insensitive compared to more complete genomic analyses.

6. PLOS authors have the option to publish the peer review history of their article (what does this mean?). If published, this will include your full peer review and any attached files.

Reviewer #1: **Yes: **Muhammad Umar Sohail

Reviewer #2: No

---

## [Author Response · Author response to Decision Letter 0]

15 Feb 2023

Effects of fenbendazole on fecal microbiome in BPH/5 mice, a model of hypertension and obesity

Beckers et al

Thank you for reviewers’ comments, see red lined below. We have rearranged the discussion per editor’s suggestion to better link results to discussion points. 

Reviewer #1: Abstract: line 42, define FBZ.

Thank you, we have corrected the mistake and added the definition on line 31. 

Please clearly mention which comparisons (gender, medication, before/after medication, obesity) are you targeting. Each result line in mentioned for a different comparison. Please give

A sentence was added to the abstract line 36-38 to aid in clarification. The main purpose of the study was to analyze fecal microbiome communities before and after medication, but it was noted that the alteration were made in a sex dependent manner. 

Give Bray-Curtis dissimilarity p value. The difference is only between male and female which means treatment has no effect. Please give each p values. Results are not coherent with the conclusions.

P values have been added to the abstract and results edited, lines 39-44

I wouldn’t say Lactobacillus as a marker of healthy gut. Lactobacillus at genus level has more complex role.

Removed from the abstract 

Methods:

What we mean by Adult (8 weeks-6 months of age)?

Thank you, verbiage was added to clarify. 

Line 86 to 91: break in smaller sentences.

Corrected, thank you. 

Line 102, 103: were not are

Corrected. 

Line 107 : exiting the room, respectively

Added, thank you.

Line 109: Samples collection:

Corrected, thank you. 

Line 110: individually in sterile empty cages

Corrected, thank you.

Line 101: were placed in sterile tubes

Corrected, thank you.

Line 129: Microbial community

Corrected, thank you.

Changes at the phyla level in BPH/5 mice with Fenbendazole treatment

Herein just mention treatment effects (FBZ). Or if you want to add gender as well, make a crystal clear difference from the start.

Only mention changes with p value < 0.05 or elsewhere mention p value. Result sentences are not written properly, and each sentence has multiple results with no clear reflection. Sometimes past and present are mixed (164-165).

P values have been added to results and result section has been edited to make things clear. 

Either figures are not clear, or p values are not mentioned. For figure 2. C, D, and E, use the same headings. As I can see that the post-male has larger bell size, this may lead to non-significant results. Lines 181 to 184: reflect that the effect is due to treatment which contradicts with the conclusions. Overall, I believe that we shall rewrite results with more coherent text and figures.

Thank you for pointing out this flaw. Titles are corrected on all figures to the same headings and p-values have been added to each figure as well. 

179: dissimilarity

Corrected, thank you.

Discussion:

Line 203: What we mean by institution?

The intended meaning was location of where those mice were housed. Corrected in text, thank you. 

Figure 2 A and B; also use the similar titles. In PCoA I can see that post treatment clustering is poor, which needs to be justified.

Titles have been edited to be similar. Clustering is poor due to the hypothesis that the treatment affected the males more greatly cause a greater community change in the post treatment samples. 

Reviewer #2: You should acknowledge as a limitation that although you found some differences in microbial frequency, you did not evaluate effects on blood pressure or obesity, which are key features of the animal model you studied. Also, 16S rRNA testing is relatively insensitive compared to more complete genomic analyses.

Thank you for the input. This is a good point and has been added to the discussion on lines 240-243.

---

## [Decision Letter · Decision Letter 1]

3 Apr 2023

PONE-D-23-00107R1Effects of fenbendazole on fecal microbiome in BPH/5 mice, a model of hypertension and obesity, a brief reportPLOS ONE

Dear Dr. Sones,

Thank you for submitting your manuscript to PLOS ONE. After careful consideration, we feel that it has merit but does not fully meet PLOS ONE’s publication criteria as it currently stands. Therefore, we invite you to submit a revised version of the manuscript that addresses the points raised during the review process.

We look forward to receiving your revised manuscript.

Kind regards,

Christopher Staley, Ph.D.

Academic Editor

PLOS ONE

Journal Requirements:

Additional Editor Comments :

Following review by two new reviewers, additional minor clarifications are requested. Importantly, sex differences and confounding factors should be accounted for where possible in the analyses, as suggested by the reviewer. Where this is not possible, it should be explained and explored in the text.

Reviewers' comments:

Reviewer's Responses to Questions

**Comments to the Author**

1. If the authors have adequately addressed your comments raised in a previous round of review and you feel that this manuscript is now acceptable for publication, you may indicate that here to bypass the “Comments to the Author” section, enter your conflict of interest statement in the “Confidential to Editor” section, and submit your "Accept" recommendation.

Reviewer #3: All comments have been addressed

Reviewer #4: (No Response)

2. Is the manuscript technically sound, and do the data support the conclusions?

Reviewer #3: Yes

Reviewer #4: No

3. Has the statistical analysis been performed appropriately and rigorously? 

Reviewer #3: Yes

Reviewer #4: No

4. Have the authors made all data underlying the findings in their manuscript fully available?

Reviewer #3: Yes

Reviewer #4: Yes

5. Is the manuscript presented in an intelligible fashion and written in standard English?

Reviewer #3: Yes

Reviewer #4: Yes

6. Review Comments to the Author

Reviewer #3: The manuscript by Beckers et al. addresses the effect of FBZ anti-parasitic treatment on stool microbiota in a mouse model of hypertension and obesity. This manuscript is well written and the analyses are straightforward. I reviewed the R1 version of the manuscript which was a response to the comments from 2 other reviewers. My comments do not take into account these earlier comments and I leave it to the editor which of my comments should be addressed. I do have a few comments:

One interesting aspect was not addressed in the discussion. These mice are inbred and are housed and fed in a very controlled environment. Therefore, I would expect (although I am not very experienced in the laboratory mouse stool microbiome) that inter-individual differences are not as high as presented in Fig. 1. If the number of words allows, I would like to see some comments on that in the discussion.

Another interesting aspect that was not clearly addressed. This is longitudinal data. The analyses can make use of these differences before and after treatment for each animal. That can be done for alphas as well as single genera. Was this done in the two way ANOVA analyses? Please comment on this.

Third interesting aspect that was not addressed. Phylum Verrucomicrobiota was higher in males after treatment which was not observed in females. The differential abundance analyses at genus-level did not show what genera were responsible for this. Please discuss why. Further, in humans Verrucomicrobiota (especially genus Akkermansia) are known to be involved in glucose metabolism and obesity. Also, genus Akkermansia is higher in females. This might be interesting in combination with the phenotype of these mice. If the number of words allows, I would like to see some comments on that in the discussion.

Analyses have been performed either at phylum-level or at genus-level, if I am correct. It is not clear how taxa were aggregated to these higher levels and it is not clear at what levels the analyses were performed, i.e., Bray-Curtis dissimilarity and differential abundances. Also, in lines 188 and 189, ASV-level analyses were mentioned but ASVs are basically at strain-level. Please make clear.

Line 115, please indicate if bead beating was part of the protocol.

Line 125, the protocol in the reference is not the manufacturer’s guideline. Illumina has another protocol. Please make sure the correct reference is given.

In line 140 it is stated that normality was confirmed, but most ASVs are not normally distributed. Please explain.

In line 141, I assume that “both p values and FDR-corrected p values <0.05 were considered significant”.

Throughout the text, taxonomic levels are written in plural form which is not always correct, e.g., phyla instead of phylum (lines 146, 149) and genera instead of genus (line 234).

Sentence in line 138 is unclear.

Reviewer #4: In this article, the authors investigated the effects of Fenbendazole on the gut microbiome of mice with hypertension and reported the difference in gut microbiome responses to the treatment between male and female mice.

This article reported an observation without controls over baseline status or any confounding factors, such as BMI, in female and male mice. The main conclusion “there are changes in the gut microbial communities after FBZ use in a sex dependent manner” is not very convincing with current study design or evidence present in the article.

Major issues:

1. Difference between baseline microbiome in male and female mice

According to figure2, there are sex-related differences at baseline. Microbiome components interact between each other. The observation of change in one single component might have multiple components involved. Plus the regression toward the mean effects in statistics. If the baseline microbiome was different between sex, then we will expect the change will be different. The authors need to calibrate for the baseline difference.

2. Lack of controls over confounding factors

As mentioned by the authors, obesity has a great influence on gut microbiome and the treatment may have sex-dependent effects on obesity. Is there a difference in BMI between female and male mice at baseline? Is there a difference in the change of BMI between female and male mice during treatment? These were not reported.

3. Statistical analysis

This study collected fecal samples at two time points from each mice. Using baseline microbiome as reference and comparing within subject changes will be a better choice than current methods, for example difference in difference model. The authors may also consider propensity score matching to remove effects of confounding factors.

4. More details in “Changes at the ASV level with Fenbendazole treatment and phenotypic outcomes”

“Changes at the ASV level with Fenbendazole treatment and phenotypic outcomes” worth more emphasis than “Changes at the phyla level in BPH/5 mice with Fenbendazole treatment”. ASV level is more informative than phyla level in showing microbiome functions.

Minor issues:

1. “Treatment had a significant effect on community composition using PERMANOVA of the Bray-Curtis dissimilarity matrix, with both BPH/5 female and male microbial communities differing post-treatment but not different by sex or interaction (treatment p=0.002, sex p=0.3, interaction p=0.05) (Figure 3b).” For “ sex p=0.3”, what was compared? Female and male samples with both time points pooled together? This analysis is actually showing that treatment had much bigger effects than sex, not supporting the conclusion.

2. The resolution of figures is too low.

7. PLOS authors have the option to publish the peer review history of their article (what does this mean?). If published, this will include your full peer review and any attached files.

Reviewer #3: No

Reviewer #4: No

---

## [Author Response · Author response to Decision Letter 1]

8 May 2023

Effects of fenbendazole on fecal microbiome in BPH/5 mice, a model of hypertension and obesity

Beckers et al

Thank you for reviewers’ comments, see red lined below. We have rearranged the discussion per editor’s suggestion to better link results to discussion points. 

Reviewer #3: The manuscript by Beckers et al. addresses the effect of FBZ anti-parasitic treatment on stool microbiota in a mouse model of hypertension and obesity. This manuscript is well written and the analyses are straightforward. I reviewed the R1 version of the manuscript which was a response to the comments from 2 other reviewers. My comments do not take into account these earlier comments and I leave it to the editor which of my comments should be addressed. I do have a few comments:

One interesting aspect was not addressed in the discussion. These mice are inbred and are housed and fed in a very controlled environment. Therefore, I would expect (although I am not very experienced in the laboratory mouse stool microbiome) that inter-individual differences are not as high as presented in Fig. 1. If the number of words allows, I would like to see some comments on that in the discussion.

Even though these mice are inbred and were housed in the same environment, some individual variation is noted on the bar graph of Figure 1, but nothing was significantly different when the individuals were used as the random effect in the 2-way ANOVA. The only thing to note of significance was the phyla Actinobacteria and Verrucomicrobia when assessing the time points pre and post, which is presented in Figure 2. 

Another interesting aspect that was not clearly addressed. This is longitudinal data. The analyses can make use of these differences before and after treatment for each animal. That can be done for alphas as well as single genera. Was this done in the two way ANOVA analyses? Please comment on this.

A 2-way ANOVA was performed with time, sex, time x sex as the fixed effects and each mouse as the random effect. This was followed up with Post hoc and Fisher’s LSD for additional analysis. A clarification was added to the statistics section within the methods to further explain. Thank you. Please see line 139-142. 

Third interesting aspect that was not addressed. Phylum Verrucomicrobiota was higher in males after treatment which was not observed in females. The differential abundance analyses at genus-level did not show what genera were responsible for this. Please discuss why. Further, in humans Verrucomicrobiota (especially genus Akkermansia) are known to be involved in glucose metabolism and obesity. Also, genus Akkermansia is higher in females. This might be interesting in combination with the phenotype of these mice. If the number of words allows, I would like to see some comments on that in the discussion.

Akkermansia was the genus of Verrucomicrobiota found in this study. This information was added to the discussion lines 232-235. Thank you. 

Analyses have been performed either at phylum-level or at genus-level, if I am correct. It is not clear how taxa were aggregated to these higher levels and it is not clear at what levels the analyses were performed, i.e., Bray-Curtis dissimilarity and differential abundances. Also, in lines 188 and 189, ASV-level analyses were mentioned but ASVs are basically at strain-level. Please make clear.

Thank you for identifying this confusion, it was clarified within the methods section. 

Line 115, please indicate if bead beating was part of the protocol.

A clarification has been added on line 118, thank you. 

Line 125, the protocol in the reference is not the manufacturer’s guideline. Illumina has another protocol. Please make sure the correct reference is given.

An additional reference has been added. 

In line 140 it is stated that normality was confirmed, but most ASVs are not normally distributed. Please explain.

This statement was removed, it was in regard to additional data that was previously removed upon last reviewers request. 

In line 141, I assume that “both p values and FDR-corrected p values <0.05 were considered significant”.

The statement within the methods was revised. 

Throughout the text, taxonomic levels are written in plural form which is not always correct, e.g., phyla instead of phylum (lines 146, 149) and genera instead of genus (line 234).

These grammatical errors were corrected. 

Sentence in line 138 is unclear.

This sentence was edited to make it clearer. 

Reviewer #4: In this article, the authors investigated the effects of Fenbendazole on the gut microbiome of mice with hypertension and reported the difference in gut microbiome responses to the treatment between male and female mice.

This article reported an observation without controls over baseline status or any confounding factors, such as BMI, in female and male mice. The main conclusion “there are changes in the gut microbial communities after FBZ use in a sex dependent manner” is not very convincing with current study design or evidence present in the article.

Major issues:

1. Difference between baseline microbiome in male and female mice

According to figure2, there are sex-related differences at baseline. Microbiome components interact between each other. The observation of change in one single component might have multiple components involved. Plus the regression toward the mean effects in statistics. If the baseline microbiome was different between sex, then we will expect the change will be different. The authors need to calibrate for the baseline difference.

A clarification and a previous references were added to note that there were no difference between male and females prior to treatment. The main difference observed was in Verrucomicrobia post treatment specifically Akkermansia, which is clarified in the discussion. 

2. Lack of controls over confounding factors

As mentioned by the authors, obesity has a great influence on gut microbiome and the treatment may have sex-dependent effects on obesity. Is there a difference in BMI between female and male mice at baseline? Is there a difference in the change of BMI between female and male mice during treatment? These were not reported.

One of the reviewers on the initial submission requested that we remove this information. Therefore, we will keep it omitted at this time. 

3. Statistical analysis

This study collected fecal samples at two time points from each mice. Using baseline microbiome as reference and comparing within subject changes will be a better choice than current methods, for example difference in difference model. The authors may also consider propensity score matching to remove effects of confounding factors.

Based on comments from reviewer 3, we used a mixed ANOVA with repeated measures for the analysis, no difference was found between sex pre-treatment. 

4. More details in “Changes at the ASV level with Fenbendazole treatment and phenotypic outcomes”

“Changes at the ASV level with Fenbendazole treatment and phenotypic outcomes” worth more emphasis than “Changes at the phyla level in BPH/5 mice with Fenbendazole treatment”. ASV level is more informative than phyla level in showing microbiome functions.

More detail was added to this section of the results, specifically the genera involved in the changes in Actinobacteria and Verrucomicrobia. 

Minor issues:

1. “Treatment had a significant effect on community composition using PERMANOVA of the Bray-Curtis dissimilarity matrix, with both BPH/5 female and male microbial communities differing post-treatment but not different by sex or interaction (treatment p=0.002, sex p=0.3, interaction p=0.05) (Figure 3b).” For “ sex p=0.3”, what was compared? Female and male samples with both time points pooled together? This analysis is actually showing that treatment had much bigger effects than sex, not supporting the conclusion.

This was clarified in the last paragraph of the discussion, lines 271-273. 

2. The resolution of figures is too low.

---

## [Decision Letter · Decision Letter 2]

31 May 2023

Effects of fenbendazole on fecal microbiome in BPH/5 mice, a model of hypertension and obesity, a brief report

PONE-D-23-00107R2

Dear Dr. Sones,

We’re pleased to inform you that your manuscript has been judged scientifically suitable for publication and will be formally accepted for publication once it meets all outstanding technical requirements.

Kind regards,

Christopher Staley, Ph.D.

Academic Editor

PLOS ONE

Additional Editor Comments (optional):

Reviewers' comments:

Reviewer's Responses to Questions

**Comments to the Author**

1. If the authors have adequately addressed your comments raised in a previous round of review and you feel that this manuscript is now acceptable for publication, you may indicate that here to bypass the “Comments to the Author” section, enter your conflict of interest statement in the “Confidential to Editor” section, and submit your "Accept" recommendation.

Reviewer #3: All comments have been addressed

Reviewer #4: All comments have been addressed

2. Is the manuscript technically sound, and do the data support the conclusions?

Reviewer #3: Yes

Reviewer #4: Yes

3. Has the statistical analysis been performed appropriately and rigorously? 

Reviewer #3: Yes

Reviewer #4: Yes

4. Have the authors made all data underlying the findings in their manuscript fully available?

Reviewer #3: Yes

Reviewer #4: Yes

5. Is the manuscript presented in an intelligible fashion and written in standard English?

Reviewer #3: Yes

Reviewer #4: Yes

6. Review Comments to the Author

Reviewer #3: My comments have been addressed properly, therefore, I recommend to accept this version of the manuscript.

Reviewer #4: The authors have explained thoroughly for the confusions and all the comments have been addressed. Thank you!

7. PLOS authors have the option to publish the peer review history of their article (what does this mean?). If published, this will include your full peer review and any attached files.

Reviewer #3: No

Reviewer #4: No

---

## [Editor Report · Acceptance letter]

2 Jun 2023

PONE-D-23-00107R2 

Effects of fenbendazole on fecal microbiome in BPH/5 mice, a model of hypertension and obesity, a brief report 

Dear Dr. Sones:

I'm pleased to inform you that your manuscript has been deemed suitable for publication in PLOS ONE. Congratulations! Your manuscript is now with our production department. 

Kind regards, 

on behalf of

Dr. Christopher Staley 

Academic Editor

PLOS ONE